# Bone Dimensional Change Following Immediate Implant Placement in Posterior Teeth with CBCT: A 6-Month Prospective Clinical Study

**DOI:** 10.3390/molecules27030608

**Published:** 2022-01-18

**Authors:** Witchayani Bungthong, Parinya Amornsettachai, Penporn Luangchana, Boontharika Chuenjitkuntaworn, Suphachai Suphangul

**Affiliations:** 1Department of Advanced General Dentistry, Faculty of Dentistry, Mahidol University, Yothi Road, Ratchathewi District, Bangkok 10400, Thailand; capuririn@gmail.com (W.B.); puay44@yahoo.com (P.A.); boontharika_c@hotmail.com (B.C.); 2Department of Oral and Maxillofacial Radiology, Faculty of Dentistry, Mahidol University, Bangkok 10400, Thailand; jee.ji@hotmail.com

**Keywords:** immediate implant placement, dental implant, xenograft, bone dimensional change, bone regeneration, cone-beam computed tomography, clinical study

## Abstract

This prospective clinical study aimed to evaluate the peri-implant hard tissue dimensional change at 6 months of immediate implant placement with bone graft materials in the posterior area using cone-beam computed tomography (CBCT). Twelve dental implants were placed concurrently following tooth extraction in the posterior area and filled with xenograft particles. The CBCT images were taken immediately after surgical procedures and then at 6 months follow-up. To evaluate the hard tissue changes, the vertical and horizontal bone thickness were analyzed and measured using ImageJ software. Paired *t*-test or Wilcoxon match-pair signed-rank test was done to analyze the changes of hard tissue values at the same level between immediately and 6 months following immediate implant placement. Independent *t*-test or Mann–Whitney U test was used to analyze the dimensional change in the vertical and horizontal direction in buccal and lingual aspects. The level of significance was set at *p* value = 0.05. All implants were successfully osseointegrated. At 6 months follow-up, the vertical bone change at the buccal aspect was −0.69 mm and at the lingual aspect −0.39 mm. For horizontal bone thickness, the bone dimensional changes at 0, 1, 5, and 9 mm levels from the implant platform were −0.62 mm, −0.70 mm, −0.24 mm, and −0.22 mm, respectively. A significant bone reduction was observed in all measurement levels during the 6 months after implant placement (*p* value < 0.05). It was noted that even with bone grafting, a decrease in bone thickness was seen following the immediate implant placement. Therefore, this technique can be an alternative method to place the implant in the posterior area.

## 1. Introduction

Significant alveolar bone resorption occurs after the extraction of the tooth [1,2,3]. The loss of bundle bone will cause an alveolar ridge alteration in both vertical and horizontal directions during the healing process [1,2,4]. At 3 months following tooth extraction in the lower posterior area, the buccal alveolar bone height was located 2.2 mm apical to the height of the lingual alveolar wall [5].

To prevent bone resorption following the extraction, one method is the immediate placement of the dental implant into the fresh extraction socket [6,7,8]. The rationale behind this is a close adaptation of the dental implant to the socket wall can reduce bone resorption [9,10]. Immediate placement of dental implant following a tooth extraction is a popular technique due to various advantages, such as reduced numbers of surgical procedures and preservation of the alveolar ridge and gingival contour [11]. However, studies have shown that this method without any grafting materials cannot inhibit bone alterations [12,13,14,15,16]. The alveolar ridge alterations are seen after 4 months following immediate implant placement [17].

Various graft materials are used to fill the extraction socket during immediate implant procedures to preserve the ridge [18,19,20,21]. The ridge preservation procedures have been used by various clinicians and researchers and showed reduced ridge alterations compared to tooth extraction without bone grafts [22,23,24]. To analyze the alveolar bone thickness, there are various methods, including direct measurement by the periodontal probe, digital caliper, and histomorphometric and radiographic analysis [25]. Among these methods, cone beam computed tomography (CBCT) is extensively used nowadays as it has high diagnostic accuracy and provides high-resolution 3D images with lower radiation doses compared to conventional CT [26]. CBCT helps to evaluate the alveolar bone width, bone dimensions (buccal and lingual bone thickness), and distance from the vital structures.

In a retrospective study [27] following immediate implant placement in maxillary anterior teeth, CBCT images showed the vertical bone height reduction was 0.82 mm while the horizontal facial bone width alterations were significantly different at the implant platform level. However, there is a lack of evidence on alveolar bone dimensional changes following immediate placement of the dental implant in the extraction socket of posterior teeth. Therefore, this prospective clinical study aimed to evaluate the peri-implant hard tissue dimensional change at 6 months after immediate placement of dental implant with bone graft materials in posterior teeth using CBCT.

## 2. Results

The T-1 (immediately after implant placement) and T-2 (6-months after implant placement) CBCT images of 12 dental implants were evaluated by one calibrated examiner. The mean values of different alveolar bone thickness between T-1 and T-2 in vertical and each horizontal measurement level were presented in Table 1 and Figure 1. When comparing the bone dimensions at each level between T-1 and T-2, significant bone changes were found in all measurement levels (*p* < 0.05) (Table 1).

The mean vertical bone height changes were −0.69 ± 0.46 mm in the buccal aspect and −0.39 ± 0.30 mm in the lingual/oral aspect. At the horizontal ridge alterations at 0, 1, 5, and 9 mm to the implant platform, the mean buccal horizontal bone thickness (H-BT) changes were −0.46 ± 0.32, −0.54 ± 0.47, −0.30 ± 0.18, and −0.18 ± 0.28 mm, respectively. Similarly, the mean oral horizontal bone thickness (H-OT) changes were −0.20 ± 0.22, −0.21 ± 0.16, −0.12 ± 0.08, and −0.12 ± 0.08 mm, respectively (Table 2). It showed no significant differences between buccal and oral aspects in vertical bone height reduction (*p* value = 0.056), as shown in Table 2. However, for the horizontal bone width alterations, significant differences were found at 0, 1, and 5 mm to the implant platform (*p* value < 0.05), but at 9 mm to the implant platform; no significant difference was seen between buccal and oral aspects (*p* = 0.795) as shown in Table 2. In addition, we found that there were no differences between the peri-implants bone loss in maxillary and mandibula arch.

## 3. Discussion

Dental implants are widely used in prosthetic rehabilitation [28,29,30,31]. Immediate implant placement without bone grafting in the posterior jaw yields a significant horizontal ridge reduction and minor mucosal recession [3,32]. Hence, we should consider augmentation at the time of implant placement. The results in this clinical study showed dimensional alterations of bone following immediate implant placement in posterior extraction sockets filled with xenograft particles. At 6 months after implant placement, dimensional bone changes were observed in all measurement levels. The results were in agreement with the results of previous immediate implant placement studies [17,25,27].

Significant bone reduction occurs after tooth extraction in both the maxillary and mandibular arch. Van der Weijden et al. [33] found that alveolar ridge width reduction was 3.87 mm (mean alveolar bone height loss was 1.67 mm). Considering this, various alveolar ridge preservation methods are proposed to maintain the ridge dimensions. Immediate implant placement is one method to prevent bone resorption. However, there are fewer scientific studies to support the prevention of bone alterations by immediate implant placement [12,13,14,15,16]. In addition, ridge preservation procedures have been studied by several authors, showing reduced ridge alterations compared to tooth extraction alone [22,23,24]. Barone et al. [34] did a randomized control trial done on the ridge alteration during the 7 months after tooth extraction using xenograft and collagen membrane. They found that there were significant differences in final horizontal and vertical bone dimensions between the ridge-preservation group and the control group. Therefore, the ridge preservation procedures using xenograft materials can reduce the bone dimensional changes.

In this present study, we analyzed the mean vertical bone height changes and horizontal ridge alterations in buccal and lingual/oral aspects at each measurement level using CBCT. The horizontal ridge changes between buccal and oral aspects at various measurement levels showed significant differences except at the 9 mm level. In our opinion and clinical observation, this reason can be due to two reasons. Firstly, the implant platform was placed 3-4 mm deeper than the free gingival margin and slightly to orally/palatally/lingually which can preserve bone height buccally. Hence, this might have resulted no difference in oral/palatal/lingual height. Secondly, this is a positive effect of Xenograft that maintains height, but for the horizontal has a mild change when compared to previous studies that are done with immediate implant placement without any graft. These results are in agreement with a previous immediate implant study [17], where they found the dimensional ridge alterations mostly notable at the buccal aspects [25]. In addition, they found at 4 months follow-up, 50% of buccal ridge resorption and 30% of lingual/palatal ridge resorption in horizontal dimensions. Hard tissue alterations occur due to the remodeling process of alveolar bone. Extensive resorption is a common situation that occurred following tooth extraction even in intact buccal plate areas [35]. This is due to the reduction of blood supply to the hard tissue following tooth extraction especially in the buccal aspect that contains cortical bone without medullary components.

Similarly, for the vertical bone height changes, there were no significant differences between the buccal and oral aspects in vertical bone height reduction. This can be due to the flapless technique used in immediate implant placement. Conceptually, the flapless technique will have less disturbance for the bone’s vascularity for vertical bone height maintenance [36].

Roe et al. [27] did a retrospective study on immediate implant placement with xenograft materials in upper anterior teeth using CBCT and focused on the facial bone dimensional alterations. They found that the vertical bone change was 0.82 mm and the horizontal bone loss, which mostly changed at the platform level, was 1.23 mm. The results of the pattern of facial bone alterations in this study are related to our study, which is performed in posterior teeth.

A study by Botticelli et al. [17] performed immediate placement of dental implant without any grafting materials and showed the horizontal resorption of the buccal alveolar ridge was at least 1.9 ± 0.9 mm and at the oral aspect was 0.9 ± 0.6 mm. Hence, in our study also the immediate placement of dental implant with bone graft materials did not prevent the peri-implant bone remodeling.

At present, various polymeric materials have been applied for the tissue engineering and regeneration of tissues including bone by reducing osteoclastic activity [37,38,39]. Various natural polymers such as alginate, cellulose, chitosan, gelatin, fibrin, collagen, laminin, decellularized extracellular matrix, and hyaluronic acid, as well as synthetic polymers including polylactic acid, polycaprolactone, polyglycolic acid, poly (ethylene glycol), and Zwitterionic polymers, can be used for tissue regeneration [37,40]. The implant biomaterials can be immunomodulatory rather than inert materials. In addition, epigenetics can act as the next generation of advanced treatment tools for future regenerative techniques [41].

In our study, four cases had small chronic apical lesions. Following extraction, sockets were cleaned with a curette, and all granulation tissue was removed and rinsed with 0.12% chlorhexidine and normal saline. The small apical lesion does not affect hard tissue dimension change.

The limitation of this prospective clinical study is the limited number of subjects. The conclusions may be influenced by study design, sample size, follow-up time, and other factors. This study can be extended as a long-term randomized-controlled clinical trial study with a larger sample size and long-term follow-up.

## 4. Materials and Methods

Ethical approval for this prospective clinical study was obtained from the Institutional Review Board of Faculty of Dentistry/Faculty of Pharmacy Mahidol University (MU-DT/PY-IRB2018/035.2106).

### 4.1. Subject Selection

The subject inclusion criteria were ≥18 years healthy subjects (medical conditions class I and II according to the American Society of Anesthesiologists classification) who needed immediate implant placement in the posterior tooth (premolar or molar) due to endodontic treatment failure, trauma, untreatable caries, or root fracture. The exclusion criteria were patients with heavy smoking (≥10 packs/day), taking immunosuppressive drugs or oral/IV bisphosphonates, history of radiotherapy/chemotherapy, pregnancy, or lactation. Teeth extracted due to periodontal problems with clinical or radiographic signs with contraindications to immediate implant placement, or absence of buccal plate were excluded from the study. All selected subjects were informed regarding the research procedures, and details on the immediate placement of dental implant and consents were obtained.

### 4.2. Research Procedure

This clinical research was divided into three phases: pre-operative, operative, and post-operative phase.

#### 4.2.1. Pre-Operative Phase

In each subject, an intraoral examination was done followed by a periapical radiograph and CBCT scan to evaluate bone quantity and quality. The preliminary impression was taken for proper evaluation for immediate implant placement. Then, each subject was appointed for the operative phase.

#### 4.2.2. Operative Phase

Tooth extraction and implant placement were performed concurrently under the local anesthesia (4% articaine containing 1:100,000 epinephrine). Tooth extraction was done with minimal traumatic technique, then, preparation of the implant bed in the peri-implant socket and implants placements were performed. The reasons for extraction were crown-root fracture due to accident (six teeth), endodontic treatment failure (four teeth), root fracture (one tooth), and dental caries exposed pulp (one tooth). One experienced oral surgeon performed all the surgical procedures, and one experienced prosthodontist performed all prosthetic procedures.

Twelve dental implants (Straumann^®®^ SLActive^®®^ bone level tapered implant, Straumann^®®^, Switzerland) were placed in seven subjects with a mean age of 43 years old (range: 29–73 years) following the manufacturer’s recommendation. Four implants were placed in the maxilla (three premolars and one molar) and eight in the mandible (eight molars).

The primary stability of dental implants was recorded and was optimal. Following the placement of the implant fixtures, deproteinized bovine bone mineral (DBBM) of particles grain size 0.5–1.0 mm (Cerabone^®®^, botiss, Germany) were placed into the gap between implant fixture and extraction socket and inserted into the customized healing abutment (Variobase, Straumann^®®^, Switzerland and Protemp^™^ 4, 3M ESPE, US and Filtek^™^ Z350 XT flowable composite, 3M ESPE, US) (Figure 2). Then, a CBCT scan (Sirona^®®^ Galileos, Germany.) was taken with a small field of view (FOV) of 5 × 5 cm^2^ after the surgical procedure as a baseline image.

#### 4.2.3. Post-Operative Phase

Post-operative pain and inflammation were controlled with the prescriptions of amoxicillin (500 mg 3 times per day for 7 days) or clindamycin (300 mg three times per day for 7 days); and for subjects with amoxicillin hypersensitivity, ibuprofen (400 mg three times per day) and acetaminophen (500 mg as pain arises).

All subjects were recommended to rinse with chlorhexidine gluconate mouthwash (0.12%) twice daily for 7 days and to perform proper oral hygiene practice with an extra-soft toothbrush in the surgical site for 4 weeks. All subjects were recalled at 1 week, 1 month (to receive the post-operative area evaluation and oral hygiene instruction), and 6 months (to take CBCT image and take an impression for definitive prosthesis). At 6-months follow-up, all dental implants were osseointegrated and were immobile and asymptomatic. Screw-cement retained single crowns were done as a final prosthesis. 

### 4.3. Outcome Measurement

All CBCT images were taken after the surgery and at 6-months after implant placement. The images were exported as DICOM files and opened with RadiAnt DICOM viewer software (version 4.6.9, Medixant) using multi-planar reconstructions for bone evaluation. In the axial view, the image was rotated until the vertical line (light blue line) bisected the implant in the buccolingual direction, whereas the perpendicular line (pink line) bisected the implant into mesial and distal aspects (Figure 3a). In the sagittal view, the image was rotated to place the implant’s long axis parallel to the pink line, and another perpendicular line (yellow line) was drawn at the platform level of the implant (Figure 3b). In the coronal view, the image was rotated to place the implant’s long axis parallel and bisect with the light blue line (Figure 3c). The image contrast was modulated to provide the discrimination of different tissue densities. Next, the axial view image was captured and then exported to evaluate peri-implant hard tissue thickness using the ImageJ image processing program.

The vertical line bisecting the implant long axis represented the implant length. A line perpendicular to the implant length line is drawn through the implant platform, which is the implant diameter. The body of the implant was then outlined. Horizontal lines parallel to the implant platform were located at 1, 5, and 9 mm to the implant platform (Figure 4a). Horizontal buccal bone thickness (H(n)-BT) at each level was measured on the line extending from the corresponding horizontal implant lines to the outline of buccal bone (light blue line). Horizontal oral bone thickness (H(n)-OT) at each level was measured on the line that extended from the corresponding horizontal implant lines to the outline of the oral bone (yellow line). Vertical bone height was the perpendicular distance from the most coronal point of buccal (VBH) and lingual/oral (VOH) bone to the line corresponding to the implant platform (Figure 4b). The total horizontal bone thickness at each level (H(n)-BOT) was measured on the line that extended from the outline of buccal bone to the outline of oral bone (blue line) (Figure 4c). Positive or negative values were recorded when the most coronal point was placed coronal or apical to the implant platform level. All measurement lines were calculated in millimeters.

For each subject, all measurement lines were measured after implant placement (T-1) and at 6 months of implant placement (T-2). One calibrated examiner performed all measurements three times and data collection to achieve the reliability of the measurements.

### 4.4. Statistical Analysis

Statistical analysis and methodology were reviewed by an independent statistician. The data were analyzed using Statistical Software (JASP version 0.9.2.0). Paired *t*-test or Wilcoxon match-pair signed-rank test was used to analyze the changes of hard tissue values at the same level between T-1 and T-2. Independent *t*-test or Mann–Whitney U Test was used to analyze the dimensional change of vertical bone height and horizontal bone thickness at the buccal and oral aspects. The level of significance was set at *p* value = 0.05.

## 5. Conclusions

The immediate placement of dental implants with bone graft materials did not prevent the peri-implant bone remodeling. Hence, even with bone grafting, a decrease in bone thickness was seen following the immediate implant placement. The alveolar ridge alteration was more evident buccally. Therefore, this technique can be an alternative way to the original treatment protocol to place an implant in the posterior areas.

## Figures and Tables

**Figure 1 molecules-27-00608-f001:**
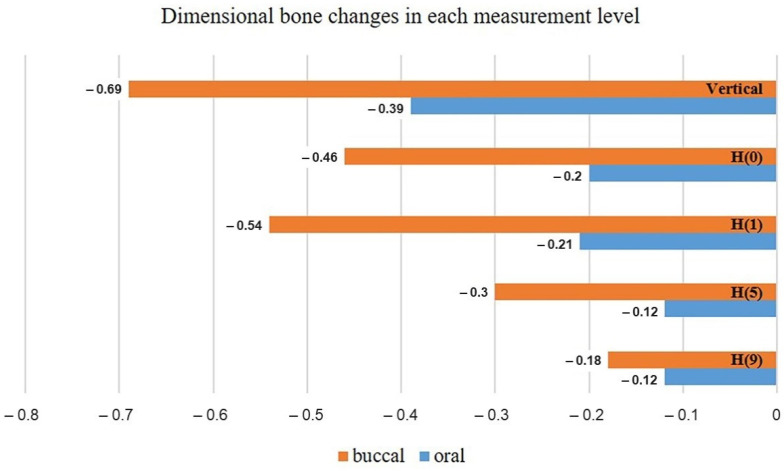
Bar chart showing the dimensional bone changes at each vertical and horizontal measurement level. (H(n) = horizontal bone changes at n millimeters to the implant platform).

**Figure 2 molecules-27-00608-f002:**
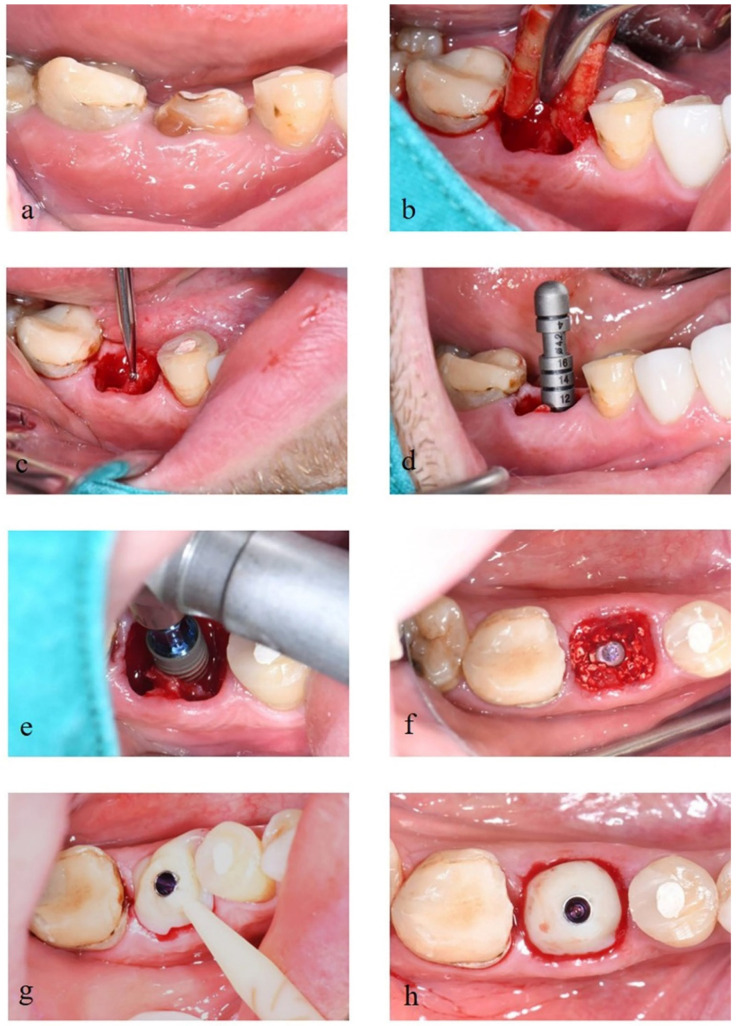
(**a**) Tooth #30 with crown-root fracture. (**b**) Tooth extraction with minimal traumatic technique. (**c**) Preparation of the implant bed using the pilot drill. (**d**) Use of depth gauge to check the implant axis and preparation depth. (**e**) Implant placement following the manufacturer’s instructions. (**f**) Placement of deproteinized bovine bone into the gap between implant fixture and socket. (**g**) Use of Variobase^®®^ and Protemp^TM^ 4 as the customized healing abutment. (**h**) Completed provisional restoration.

**Figure 3 molecules-27-00608-f003:**
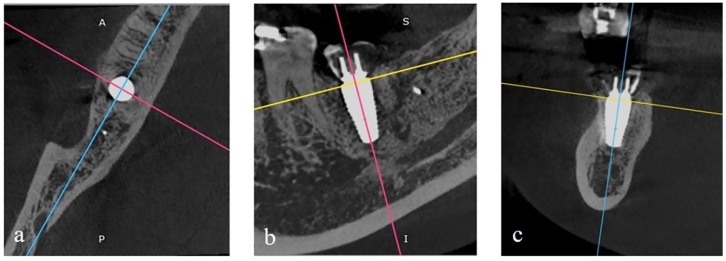
(**a**) In axial view, the vertical line (light blue line) bisects the implant in the buccolingual direction, and the perpendicular line (pink line) bisects the implant into the mesiodistal direction. (**b**) In sagittal view, rotation of the image was done until the implant’s long axis was parallel to the pink line, and the yellow line was located at the platform level of the implant. (**c**) In the coronal view, the implant’s long axis is kept parallel and bisects with the light blue line and the yellow line located at the platform level of the implant.

**Figure 4 molecules-27-00608-f004:**
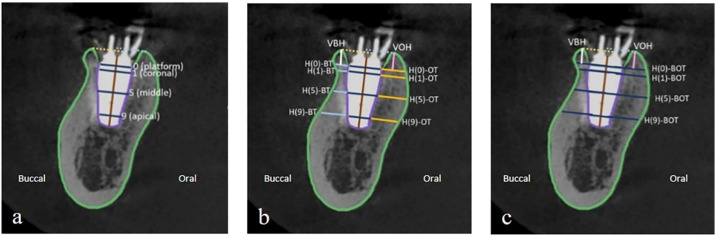
(**a**) The outline of the alveolar bone border and the body of the implant. Parallel lines to the implant platform (0) are located at 1, 5, and 9 mm to the implant platform, and the most coronal point of the buccal and oral bone. (**b**) Horizontal buccal bone thickness (H(n)-BT; light blue line) and horizontal oral bone thickness (H(n)-OT; yellow line) at each level measured on the line extended from the corresponding horizontal implant line to the outline of alveolar bone. Vertical bone height was the perpendicular distance from the most coronal point of buccal (VBH; white line) and oral bone (VOH; pink line) to the line corresponding to the implant platform. (**c**) Horizontal bone thickness at each level (H(n)-BOT; blue line) was measured on the line that extended from the outline of buccal bone to the outline of oral bone.

**Table 1 molecules-27-00608-t001:** Comparison of bone thickness between time intervals and measurement levels.

Level	Bone Dimension (mm)		Dimensional Change (mm) (T-2) − (T-1)	*p* Value
	T-1 (Mean ± SD)	T-2 (Mean ± SD)		
Vertical bone height:				
VBH	1.58 ± 0.48	0.89 ± 0.42	−0.69 ± 0.46	<0.001 *
VOH	1.47 ± 1.23	1.08 ± 1.12	−0.39 ± 0.30	0.001 *
Horizontal buccal bone thickness:				
H(0)-BT	2.60 ± 1.04	2.14 ± 1.00	−0.46 ± 0.32	<0.001 *
H(1)-BT	3.04 ± 1.12	2.5 ± 1.17	−0.54 ± 0.47	0.001 *
H(5)-BT	3.57 ± 1.38	3.27 ± 1.32	−0.30 ± 0.18	<0.001 *
H(9)-BT	4.14 ± 1.98	3.96 ± 2.07	−0.18 ± 0.28	0.014 *
Horizontal oral bone thickness:				
H(0)-OT	3.01 ± 1.79	2.82 ± 1.83	−0.20 ± 0.22	0.009 *
H(1)-OT	3.15 ± 1.90	2.94 ± 1.87	−0.21 ± 0.16	0.001 *
H(5)-OT	4.40 ± 1.78	4.28 ± 1.80	−0.12 ± 0.08	<0.001 *
H(9)-OT	4.78 ± 1.82	4.66 ± 1.82	−0.12 ± 0.08	<0.001 *
Total horizontal bone thickness:				
H(0)-BOT	10.10 ± 1.54	9.49 ± 1.59	−0.62 ± 0.43	<0.001 *
H(1)-BOT	11.07 ± 1.73	10.37 ± 1.73	−0.70 ± 0.49	<0.001 *
H(5)-BOT	12.34 ± 1.55	12.10 ± 1.66	−0.24 ± 0.21	0.003 *
H(9)-BOT	12.10 ± 1.55	11.88 ± 1.54	−0.22 ± 0.33	0.007 *

SD = standard deviation, T-1 = immediately after immediate implant placement, T-2 = 6 months following immediate implant placement, VBH = vertical buccal bone height, VOH = vertical oral bone height, H(n)BT = horizontal buccal bone thickness at n millimeters to implant platform, H(n)OT = horizontal oral bone thickness at n millimeters to implant platform, H(n)BOT =horizontal total bone thickness (from buccal to oral aspect) at n millimeters to implant platform. * Significant difference at *p* value < 0.05.

**Table 2 molecules-27-00608-t002:** Comparison of dimensional bone changes in vertical bone height and horizontal bone width between buccal and oral aspects.

Measurement Level	*p* Value (T2 − T1)
Vertical height	0.056
Horizontal thickness	
H(0)	0.024 *
H(1)	0.029 *
H(5)	<0.001 *
H(9)	0.795

T-1 = immediately after immediate implant placement, T-2 = 6 months following immediate implant placement, H(n) = horizontal bone thickness at n millimeters to implant platform. * Significant difference at *p* value < 0.05.

## Data Availability

Data are available upon request.

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
