# Peer review of "Bone Dimensional Change Following Immediate Implant Placement in Posterior Teeth with CBCT: A 6-Month Prospective Clinical Study"

_molecules, 2022, doi:10.3390/molecules27030608_

Round 1
Reviewer 1 Report
The study have a main contributions in measurements performed on CBCT images, of the peri-implant hard tissues dimensional changes referring to the vertical bone height and of horizontal bone thickness, on buccal and oral aspects.
Material and Methods section should be located before results and discussions.
I suggest that sentence located in the lines 69-79 to be moved to material and method section.
I recommend to the authors to specify in material and methods, respectively in results section:
- if this research contained a control group, without xenograft particles insertion, because in lines 282-284 the authors stated that ”The amount of bone alterations was less in implant placement with xenograft particles compared to the implant placement without grafting materials”.
- if the loaded implants supported only single-crowns;
- if there were observed differences between the peri-implants bone loss in maxillary and in mandibula;
- if there were implants failure.
The manuscript is clear, well-structured and relevant for the dental implantology field.
Only 9 references (no. 3, 7, 8, 17, 28-32) of 32 are of the last 5 years.
Because of contained data value, maybe the manuscript will be more appropriate for publication in another MDPI journal, for example Dentistry Journal, Journal of Clinical Medicine, Medicina, or Tomography.
Author Response
Response to Reviewer 1 Comments
The study have a main contributions in measurements performed on CBCT images, of the peri-implant hard tissues dimensional changes referring to the vertical bone height and of horizontal bone thickness, on buccal and oral aspects.
Thank you for your positive comments. Corrections in the Manuscript for Reviewer 1 are highlighted in Yellow color.
Material and Methods section should be located before results and discussions.
I suggest that sentence located in the lines 69-79 to be moved to material and method section.
Response: This paragraph was moved to the Materials and Methods.
I recommend to the authors to specify in material and methods, respectively in results section:
- if this research contained a control group, without xenograft particles insertion, because in lines 282-284 the authors stated that “The amount of bone alterations was less in implant placement with xenograft particles compared to the implant placement without grafting materials”.
Response: We did not have control group. We only compared before and after the treatment. We have edited in the abstract and in the conclusion section.
- if the loaded implants supported only single-crowns;
Response: Yes. In this study, we focused only single crowns implant supported.
- if there were observed differences between the peri-implants bone loss in maxillary and in mandibula.
Response: We observed that there were no differences in maxilla and mandible. This is added in results section. (Page 3)
- if there were implants failure.
Response: No implant failure in this clinical study. (Page 8)
The manuscript is clear, well-structured and relevant for the dental implantology field.
Only 9 references (no. 3, 7, 8, 17, 28-32) of 32 are of the last 5 years.
Response: The content of Section 1-5, is improved, and the redundancies are removed, focusing on main points. Recent references are added.
Because of contained data value, maybe the manuscript will be more appropriate for publication in another MDPI journal, for example Dentistry Journal, Journal of Clinical Medicine, Medicina, or Tomography.
Response: After talking with Editors, they told us that our research falls within the scope of the Special Issue “Advancements of Materials for Prosthodontics and Dental Implantology”. We are hopeful.
Reviewer 2 Report
The reviewer was interested in this well-reported manuscript. This prospective clinical study aimed to evaluate the peri-implant hard tissue dimensional change at 6 months of immediate placement of dental implant with bone graft materials in posterior teeth using CBCT. The measurement figure in the manuscript was clear and easy to follow. This study has certain clinical reference value for immediate placement of posterior teeth. However, there are several problems in the manuscript that need to be improved.
1.This study only included 7 subjects and 12 implants, and the generalisability of the conclusions may be limited.
2.“A study by Botticelli et al.performed immediate placement of dental implant without any grafting materials and the show the horizontal resorption of the buccal alveolar ridge was at least 1.9 ± 0.9 mm and at the oral aspect was 0.9 ± 0.6 mm. But in our study, we did immediate placement of dental implant with xenograft particles and found less bone resorption.”
Comparing the immediate placement of dental implant with xenograft particles group in this study with the group without any grafting materials in other studies, the conclusions may be influenced by study design, sample size, follow-up time and other factors.
3. In this study, 4 cases of root canal treatment failure were included. Is there apical infection? If there is apical infection, how do you deal with it? Will it affect hard tissue dimensional change of immediate placement of dental implant with bone graft materials in posterior teeth?
4.The data were evaluated by one calibrated examiner, and how many times did you measure? Is there a measurement bias?
5.This study found the horizontal ridge changes between buccal and oral aspects at various measurement level showed significant differences except at the 9-mm level, However, there is no significant differences between buccal and oral aspect in vertical bone height reduction. What do you think of this result?
6. Figure1 repeats part of the data in Table1, can it be simplified?
7. There are only 9 references in recent 5 years, accounting for 28%. It is recommended to read and supplement the latest literature.
Author Response
Response to Reviewer 2 Comments
The reviewer was interested in this well-reported manuscript. This prospective clinical study aimed to evaluate the peri-implant hard tissue dimensional change at 6 months of immediate placement of dental implant with bone graft materials in posterior teeth using CBCT. The measurement figure in the manuscript was clear and easy to follow. This study has certain clinical reference value for immediate placement of posterior teeth. However, there are several problems in the manuscript that need to be improved.
Thank you for your positive comments. Corrections in the Manuscript for Reviewer 2 are highlighted in Green color.
1.This study only included 7 subjects and 12 implants, and the generalisability of the conclusions may be limited.
Response: By considering study criteria and time, we did this study in only 12 implants. And we have added these in the limitation and future studies we are planning select more number of patients. (Page 5)
2.“A study by Botticelli et al. performed immediate placement of dental implant without any grafting materials and the show the horizontal resorption of the buccal alveolar ridge was at least 1.9 ± 0.9 mm and at the oral aspect was 0.9 ± 0.6 mm. But in our study, we did immediate placement of dental implant with xenograft particles and found less bone resorption.”
Response: Yes, in our study, it showed less horizontal and vertical hard tissue resorption which is positive for preserve bone dimensions after the tooth loss.
Comparing the immediate placement of dental implant with xenograft particles group in this study with the group without any grafting materials in other studies, the conclusions may be influenced by study design, sample size, follow-up time and other factors.
Response: The results in this study may influence clinical practice for use Xenograft fill the socket implant gap to reduce hard tissue resorption that will maintain natural structure for long term outcome. The conclusions may be influenced by study design, sample size, follow-up time and other factors, we have added this in limitations.
3. In this study, 4 cases of root canal treatment failure were included. Is there apical infection? If there is apical infection, how do you deal with it? Will it affect hard tissue dimensional change of immediate placement of dental implant with bone graft materials in posterior teeth?
Response: In our study, four cases had small chronic apical lesions. Following extraction, sockets were cleaned with curette and all granulation tissue were removed and rinsed with 0.12% chlorhexidine and normal saline. The small apical lesion has no effect to hard tissue dimensions change. We added this in discussion. Page 5
4. The data were evaluated by one calibrated examiner, and how many times did you measure? Is there a measurement bias?
Response: All data was analyzed with only examiner (WB). She measured 3 times. There was no bias.
5. This study found the horizontal ridge changes between buccal and oral aspects at various measurement level showed significant differences except at the 9-mm level, However, there is no significant differences between buccal and oral aspect in vertical bone height reduction. What do you think of this result?
Response: In our opinion, vertical bone height reduction result has the dominant effect of Xenograft that maintain height of the socket and surgical technique that place implant platform beneath free gingival margin may remain vertical bone height also.
6. Figure1 repeats part of the data in Table1, can it be simplified?
Response: The repeated data in Table 1 are removed.
7. There are only 9 references in recent 5 years, accounting for 28%. It is recommended to read and supplement the latest literature.
Response: We added more recent literatures.
Round 2
Reviewer 2 Report
There are several problems in the manuscript that can be improved.
1. If there is no control group without any grafting materials in the manuscript, how can conclusions “immediate placement of dental implant with xenograft particles had less bone resorption”be drawn?
2. If the data is measured 3 times by one examiner, it needs to be mentioned in the manuscript.
3. What do you think about the difference in horizontal ridge changes between buccal and oral aspects but no difference in vertical bone height?
Author Response
Response to Reviewer 2 Comments
There are several problems in the manuscript that can be improved.
Thank you for your comments. Corrections in the Manuscript for Reviewer 2 are highlighted in Green color.
1. If there is no control group without any grafting materials in the manuscript, how can conclusions “immediate placement of dental implant with xenograft particles had less bone resorption”be drawn?
The line is edited. Line 152-154.
2. If the data is measured 3 times by one examiner, it needs to be mentioned in the manuscript.
This is added in the manuscript. Line 281-282.
3. What do you think about the difference in horizontal ridge changes between buccal and oral aspects but no difference in vertical bone height?
Previously, there was mistake placement of * at P value =0.056. Apology for this.
From the Table 2, it is found that there is no significant difference between buccal and oral aspects in vertical bone height reduction (P value =0.056) which is similar to the no difference in vertical bone height.
In our opinion and clinical observation, this reason can be due to two reasons. Firstly, the platform of implant was placed deeper free gingival margin around 3-4 mm and slightly to oral/palatal/lingual, that can preserve bone height at buccal and had changed no difference with oral/palatal/lingual height. Secondly, this is positive effect of Xenograft that maintains height, but however for horizontal has mild change when compared to previous studies that done immediate implant placement without any graft.
These are added in the discussion. Line 125-131.
Finally, English correction is done throughout the manuscript.